# Low-Molecular Pyrazine-Based DNA Binders: Physicochemical and Antimicrobial Properties

**DOI:** 10.3390/molecules27123704

**Published:** 2022-06-09

**Authors:** Paulina Mech-Warda, Artur Giełdoń, Anna Kawiak, Natalia Maciejewska, Mateusz Olszewski, Mariusz Makowski, Agnieszka Chylewska

**Affiliations:** 1Department of Bioinorganic Chemistry, Faculty of Chemistry, University of Gdańsk, Wita Stwosza 63, 80-308 Gdańsk, Poland; paulina.mech@phdstud.ug.edu.pl (P.M.-W.); mariusz.makowski@ug.edu.pl (M.M.); 2Department of Theoretical Chemistry, Faculty of Chemistry, University of Gdańsk, Wita Stwosza 63, 80-308 Gdańsk, Poland; artur.gieldon@ug.edu.pl; 3Institute of Biotechnology, Intercollegiate Faculty of Biotechnology, University of Gdańsk and Medical University of Gdańsk, Abrahama 58, 80-307 Gdańsk, Poland; anna.kawiak@biotech.ug.edu.pl; 4Faculty of Chemistry, Gdańsk University of Technology, Gabriela Narutowicza 11/12, 80-233 Gdańsk, Poland; natalia.maciejewska@pg.edu.pl (N.M.); mateusz.olszewski@pg.edu.pl (M.O.)

**Keywords:** chloride effect, DNA-binding, pyrazine-based derivatives, DFT, physicochemical properties

## Abstract

Pyrazine and its derivatives are a large group of compounds that exhibit broad biological activity, the changes of which can be easily detected by a substituent effect or a change in the functional group. The present studies combined theoretical research with the density functional theory (DFT) approach (B3LYP/6-311+G**) and experimental (potentiometric and spectrophotometric) analysis for a thorough understanding of the structure of chlorohydrazinopyrazine, its physicochemical and cytotoxic properties, and the site and nature of interaction with DNA. The obtained results indicated that 2-chloro-3-hydrazinopyrazine (2Cl3HP) displayed the highest affinity to DNA. Cytotoxicity studies revealed that the compound did not exhibit toxicity toward human dermal keratinocytes, which supported the potential application of 2Cl3HP in clinical use. The study also attempted to establish the possible equilibria occurring in the aqueous solution and, using both theoretical and experimental methods, clearly showed the hydrophilic nature of the compound. The experimental and theoretical results of the study confirmed the quality of the compound, as well as the appropriateness of the selected set of methods for similar research.

## 1. Introduction

Biomolecules such as DNA are the major intracellular targets of drugs and a wide variety of biologically active compounds. These compounds may either bind to DNA or interfere with its vital functions. An analysis of these interactions provides insight into the effects of a compound on living organisms. The establishing of the binding mode of the small organic molecules with DNA is a promising research topic. Small molecules can bind with DNA through covalent or noncovalent interactions [1]. Studies focusing on deciphering the interactions occurring between a compound and DNA help us to better understand the mechanism of its action at the molecular level. They allow us to predict how an active substance binds to DNA and the atoms involved, thereby enabling the preliminary determination of the interactions of structural analogs [2,3,4].

The three main modes of noncovalent interactions are groove binding, intercalary binding, and electrostatic interactions. In minor groove binding, the molecules are closely in contact with the walls of the groove, resulting in numerous hydrogen bonds. Major groove binding also involves hydrogen bond formation between the molecules and DNA. Intercalation occurs while planar aromatic ring systems are located between base pairs, which is the mode of interaction often used by planar organic molecules containing a few aromatic condensed rings to bind to DNA [5,6]. Electrostatic interactions occur between the compound and the bases and phosphate backbone of DNA (negatively charged sugar-phosphate structure). Those interaction modes can be studied by various experimental techniques as well as theoretical chemistry methods, including nuclear magnetic resonance, spectrophotometry, electrochemistry, and molecular docking [5,7].

Improper use of antibiotics has resulted in the development of microorganisms that are harmful to human health. This phenomenon, termed “drug resistance”, tends to develop rapidly, and hence, researchers have been trying to meet the demand for new-generation antimicrobials with a higher therapeutic index. Prediction of the effects of structural modifications can allow evaluating the stability, effectiveness, and destination of newly formed products. Most importantly, the acid-base properties of compounds, which are crucial in understanding their biological activity, should be determined, as well as the solubility and lipophilicity of molecules, and also indicate the pattern of their distribution in the natural environment [8,9,10,11].

Pyrazine and its derivatives are a large group of compounds that exhibit broad biological activity, the changes of which can be easily detected by a substituent effect or a change in the functional group. Hence, 2-chloro-3-hydrazinopyrazine (2Cl3HP) is considered an interesting and promising research object compared with known pyrazinamide or the previously studied 2-hydrazinopyrazine (2HP) [1]. The important aim of research on compounds is to determine the interaction site in a living organism, and thus its molecular target. These targets are mostly biomacromolecules such as proteins or DNA. When DNA is regarded as the molecular target, it is necessary to consider the binding site of the molecule, as well as the type and mode of interaction and its strength, which is described in practice as interaction constant. To the best of our knowledge, our work is the first to analyze the structure, acid-base properties, lipophilicity, and cytotoxicity of 2Cl3HP as well as its interaction with DNA. To obtain the most accurate and reliable results, we performed both theoretical and experimental research. We use theoretical methods intentionally to support our experiments and show some tendencies qualitatively in the case of acid-base properties establishing of 2Cl3HP studied. We are aware that our DFT results might not fit well in a quantitative way, but they allow us to analyze, for example, conformational hypersurface, IR spectra, relative interaction energy, and finally HOMO LUMO orbitals, to describe the electronic properties. Moreover, the use of calculations with the water model in conjunction with potentiometry and spectrophotometry will make it possible to learn about the properties of 2Cl3HP in solution: acid-base properties and lipophilicity.

Additionally, the DFT results will be compared with those acquired recently for 2-hydrazinopyrazine (2HP) [1] to determine the relationships associated with the chloride substituent effect. A known antituberculosis drug, pyrazinamide (PZA), was used for comparison, and its interaction with DNA was studied as a reference [12,13]. Many papers from recent years concern structural modifications of the mentioned drug—pyrazinamide—in the context of obtaining a biologically active substance [14,15,16]. Other derivatives were also tested by us earlier, e.g., PTCA, PAOX, ABMAP, or DPP [17,18].

## 2. Results and Discussion

2-chloro-3-hydrazinopyrazine (2Cl3HP) is an analog of 2-hydrazinopyrazine (2HP). The structures of both are shown in Figure 1.

The pyrazine ring of 2Cl3HP has two substituents: a chlorine atom in the second position and a hydrazine group in the third position. Its structure with the spatial map of the electron density distribution received in the Chem3D^®^ program by *Perkin Elmer* is shown in Figure 2.

### 2.1. Molecular Geometry

A search for the potential energy hypersurface resulted in the identification of three isomers of the studied chloride derivative of 2HP. The chlorine atom can be located in the second, fifth, or sixth position of the pyrazine ring. The minimum potential energy surface is the characteristic of a compound having a pyrazine ring substituted with a hydrazine group in the third position and a chlorine atom in the second position. The relative stability of the isomers in vacuo was determined. The optimized molecular structures of the isomers, along with their atoms numbered and their relative Gibbs free energy values, are shown in Figure 3.

The total minimum energy determined by optimizing the B3LYP/6-311+G** is −834.667217 Hartree/part.

The Gibbs free energy differences of the B and C isomers were in the range of 0.84–2.22 kcal/mol compared to the isomer with the lowest energy (A), which indicates their significant contribution to the structure of 2ClHP.

The optimized geometry parameters of 2Cl3HP, which were obtained based on B3LYP/6-311+G** calculations, together with bond lengths, angles, and selected dihedral angles, are listed in Appendix A.

### 2.2. Vibrational (IR) Spectra Analysis

The oscillation spectra of 2Cl3HP established both experimentally and theoretically were plotted as a dependence of the intensity and oscillating frequency and shown in Figure 4. Calculations were performed for a single molecule in a vacuum. The optimized geometry was at the local lowest point on the potential energy surface, as indicated by the theoretically predicted vibration spectrum, which did not have an imaginary wavenumber. The wavelength values obtained both theoretically and experimentally were qualitatively consistent. This is due to the approximate nature of the theoretical methods used and the different states of the studied compounds.

Vibrations for six-membered aromatic rings, including benzene and pyrazines, usually appear as two or three bands due to skeletal vibration, with the strongest usually found around 1500 cm^−1^. The bands observed at 1522 and 1478 cm^−1^ were assigned to C–C ring vibrations. Those identified at 1166, 1111, 1063, and 620 cm^−1^ in the infrared (IR) spectra were assigned to the C–H ring in-plane bending modes. The bands identified at 1562 and 1612 cm^−1^ were related to C–N in-ring bending vibrations, while the peaks at 965, 870, and 854 cm^−1^ were ascribed to the C–H out-of-plane vibrations. Vibrations in the region of 750–700 and 450–400 cm^−1^, respectively, are often attributed to the C–Cl group, whereas in this study the bands identified at 743 and 451 cm^−1^ were assumed to be related to this group. Heteroaromatic compounds with the amine group showed N–H stretching vibrations at the range of 3500–3220 cm^−1^. In the spectrum of 2Cl3HP, the bands at 3645 and 3527 cm^−1^ were assigned to the NH_2_ group as symmetry and asymmetry modes of vibrations, respectively. The presence of C–H vibrations in the range of 3000–3100 cm^−1^, which is the characteristic region of C–H vibrations, may indicate heteroaromatic structure. The bands found at 3195 and 3166 cm^−1^ in the spectrum of 2Cl3HP could be attributed to the C–H stretching vibrations. The C–N and N–H vibrations can be identified between 1240 and 1350 cm^−1^ as stretching vibration modes, and in this study, they were observed at 1240, 1279, 1341, and 1350 cm^−1^, respectively [19,20,21].

The IR spectra of the chloride derivative of 2HP with a chloride substituent in three different positions (2Cl3HP (black), 2-chloro-5-hydrazinepyrazine (red), and 2-chloro-6-hydrazinepyrazine (blue)), with the corresponding structures in frames, are depicted in Appendix A. The areas with the greatest differences observed in the compared spectra, namely 3200–3000, 1300–800, and 600–300 cm^−1^, are indicated by a dashed line. The highest band intensity for 2-chloro-5-hydrazinopyrazine can be seen at 500 cm^−1^, as well as the appearance of double bands, which indicates the symmetry of the compound (para position). A change in the intensity of the bands, increasing in order from black through red to blue, can be observed at 1000 cm^−1^. In addition, there is a noticeable shift of the peaks toward lower frequencies in the same order.

### 2.3. Frontier Molecular Orbitals (FMOs) Considerations and UV-Vis Spectra Analysis

As shown in Figure 4B, TD-DFT/B3LYP/6-311+G** calculations predicted an intense electronic transition at 231 nm with an oscillator strength f = 0.175 and at 304 nm with f = 0.05, in agreement with the experimental data (λexp = 241 nm, λexp = 324 nm). It has been found elsewhere [22,23] that this quite simple and cheap computational method reproduces experimental results reasonably well. This electronic absorption is described as the excitation of an electron from the highest occupied molecular orbital (HOMO) to the lowest unoccupied molecular orbital (LUMO) and corresponds to the transition from the ground to the first excited state. The HOMO and LUMO distribution with the energy gap is shown in Figure 5, with the negative and positive phases marked in grey and green, respectively. It can be noted that the HOMO orbital is spread over the whole molecule, covering both the hydrazine group and pyrazine ring, whereas the LUMO orbital is located only over the pyrazine ring. The HOMO–LUMO transition involves the transfer of electron density from the hydrazine group to the pyrazine ring, which may be related to the presence of the electronegative, strongly electron-withdrawing chlorine substituent. The calculated energy values of HOMO and LUMO in the gas phase are −6.581 and −1.780 eV. The HOMO–LUMO gap energy in vacuo is Δ = 4.800 eV. The negative values of HOMO and LUMO energies, including the large energy gap and negative chemical potential, indicate that 2Cl3HP is highly stable. In turn, it implies that 2HP has slightly higher stability, which is probably due to the presence of a chlorine atom in the molecule. Using the energy values of the frontier orbitals, other properties can be determined by the relationships found in the literature [24,25]. The parameters describing the electronic properties of the molecule are summarized in Table 1. In comparison to the previously described 2HP, the chloride derivative of 2Cl3HP shows lower ionization energy, electron affinity, and absolute hardness, but higher absolute softness and potential. The calculated properties of chloride derivatives of both 2Cl3HP and 2HP reveal that these compounds tend to capture electrons instead of donating them. The larger ionization potential of 2Cl3HP indicates its need for high energy to become a cation, which points out the easier ionization of its chloride derivative compared to that of 2HP. The strength of an acceptor molecule is measured by the electron affinity exhibited during energy release, when one electron is added to a LUMO. The acceptor should have a high electron affinity. It was noted that the addition of an electronegative chloride substituent to the ring leads to a reduction in the electron affinity of the molecule, as shown in Table 1, which compares affinity values of the studied compounds. The lower hardness and higher softness of 2Cl3HP in comparison to 2HP indicate its weaker resistance toward the deformation of electron cloud during the chemical processes and its higher polarizability.

Appendix A show the HOMO and LUMO distributions of the B and C isomers, analogically to 2Cl3HP in Figure 5 in the paper. The energy values are, respectively, HOMO: −7.096513164; LUMO: −4.11439392 and HOMO-LUMO gap: 2.98212 [eV] for isomer B and HOMO: −7.230394236; LUMO: −4.2398394 and HOMO-LUMO gap: 2.99056 [eV] for the C isomer. Based on the values of the orbital energies HOMO and LUMO and the values of the HOMO–LUMO gap (see also experimental results to compare Appendix A), it can be concluded that both isomers are of similar stability. On the other hand, based on the large difference between the HOMO–LUMO gap value of the A isomer and those for the B and C isomers, it can be inferred that the A isomer will be the most durable and the most stable, which also results from the energy analysis (Appendix A). This is an additional confirmation of the utility of the selection of the A isomer for further research.

### 2.4. Evaluation of 2Cl3HP Partition Coefficient (logP)

When considering a compound having biological effects and DNA as its cellular target, it is crucial to determine its lipophilicity, which is an important aspect of physicochemical characterization. This parameter describes the ability of a compound to be dissolved in nonpolar systems. Thus, it allows for predicting the behavior of the compound in the organism and the environment, and whether it can penetrate the cell membrane and into the cells. Moreover, lipophilicity is an indicator of the reactivity of a compound and its durability of binding with components present in the environment or cells [26]. Typically, this parameter is expressed as the partition coefficient (logP). Based on theoretical calculations, the value of logP can be derived from the equation [27,28]:(1)$$\mathrm{logP}\;=\frac{(G_{H_{2}O}-G_{n-\mathrm{oct}})\cdot 627.5095}{\mathrm{RTln}10}$$
where $G_{H_{2}O}$ is the Gibbs free energy of compound in water, G_n-oct_ is the Gibbs free energy of compound in n-octanol, R is universal gas constants and T is the temperature (298.15 K). The multiplier of 627.5095 kcal·mol^−1^·a.u.^−1^ was used to convert the units of Gibbs free energy values to kcal/mol. The logP value is dimensionless.

In this study, the logP values of 2Cl3HP and 2HP were determined in a standard n-octanol/water system.

The obtained theoretical values are shown in Table 2. The logP value was estimated to observe the tendency (a sign of value) and determine whether the compound is hydrophobic or hydrophilic. This value is determined indirectly due to the properties of n-octanol and water, and, therefore, an estimation based on the sign of the logP value is completely sufficient. It can be seen that both calculated values are negative, which indicates the hydrophilic character of 2Cl3HP. In addition, the values are all above that of 2HP [1]. A comparison analysis of logP values determined previously for 2HP (−0.468) and those found in the safety data sheet for 2Cl3HP (−0.935) suggested the introduction of a Cl atom instead of an H atom to the third position of the 2HP ring causes a change in logP value. Although both pyrazine derivatives, 2HP and 2Cl3HP, can be considered as hydrophilic molecules, the change in logP value seems to be lower for 2Cl3HP due to the presence of a chloride substituent in its structure.

However, the change of pKa values between 2HP and 2Cl3HP needs to be discussed. The Cl^−^ ion is an electron-donating substituent and makes the π system of 2Cl3HP more nucleophilic.

### 2.5. Acid-Base Properties

All proposed ionic forms present in an aqueous solution were built and optimized, and their relative Gibbs free energy values are shown in Appendix A. The reference point is the form having the lowest Gibbs free energy for a given degree of ionization. By comparing the energy values, the preferred ionic forms that are most likely to occur were identified (sites with the lowest relative energies are marked in the colored boxes in Appendix A). Using these forms, a model of acid-base equilibria occurring in an aqueous solution was proposed. As most biochemical reactions take place in the water environment and mimic the conditions inside living cells, water was chosen to determine the acid-base properties.

Based on energy analysis, a model of equilibrium occurring in an aqueous solution was proposed. Calculations were carried out using a known thermodynamic cycle that combines gas-phase deprotonation Gibbs free energies and solvation energies, a detailed description of which can be found in the literature [29,30]. The final pKa values were determined by the expression:(2)$$\mathrm{pKa}\;=\frac{\Delta G_{\left(\mathrm{aq}\right)}^{*}}{\mathrm{RTln}10}$$
where $\Delta G_{\left(\mathrm{aq}\right)}^{*}$ is the Gibbs free energy change at a standard state of 1 M.

DFT simulations and experimental analyses performed in the study confirmed the possible protonation of three atoms (nitrogen) and deprotonation of one (N(7)H (Figure 2)), and thus the existence of four acid-base equilibria, respectively. The pKa values determined both theoretically and experimentally are presented in Table 3. The values of acid dissociation constants related to DFT calculations were determined based on the proposed equilibrium in the water system presented in Table 4.

To assess pKa values, the following DFT methods were employed—PCM/B3LYP/6-311+G** and SMD/M06-2X/6-311+G**. These methods somewhat based on routine calculations and are still in common usage [31,32], allowing us to estimate pKa values qualitatively well, which is desirable in our work. It is the cheapest way to reproduce the experimental results qualitatively well and a good preliminary study of acid-base properties to estimate the tendency before carrying out an expensive experiment. The calculated values were found to exhibit a consistent trend of changes in both theoretical and experimental results. Moreover, the theoretically obtained values differed numerically from those obtained experimentally but were qualitatively consistent and indicated the same trend of changes. It seems that the differences observed in the values obtained stemmed from the approximations used in theoretical calculations with continuous solvent models. A major source of errors in the calculation of pKa values is the change in the Gibbs free energy of solvation estimated by implicit solvation models.

### 2.6. Spectroscopic and Potentiometric Results

To ensure the appropriate selection of 2Cl3HP dosage that can result in the desired therapeutic, diagnostic, or microbiological effects, an assay for establishing the pKa value was carried out. Indeed, the known acid-base profile of 2Cl3HP allowed for predicting its ionic forms that can be found in body fluids. Accordingly, pH-dependent spectrophotometric (Figure 6) as well as potentiometric titrations (Figure 6) were performed to predict the behavior of 2Cl3HP in the specific, inter-, or intracellular environment. Microtitration experiments that enabled the determination of complete acid-base properties of 2Cl3HP, based on the diversity of measurable parameters, were also planned and developed to support the obtained theoretical results. Initially, the UV spectroscopic studies of 2Cl3HP about pH changes were performed and the electronic spectra were registered. The low-wavelength part of the spectra was dominated by wide and intense bands, which are attributed to the double bonds of the pyrazine scaffold present in the structure of 2Cl3HP. The first deprotonation process of the studied 2Cl3HP system seemed to occur spontaneously, which is indicated by the initial spectrum of the 2Cl3HP solution at pH 1.64, shown in Figure 6A (black), but according to the initial acidification of the 2Cl3HP parent sample, the proton transfer of this process is related to the cationic form of this compound. The intensity of absorption peak at 225 and 315 nm decreased upon the addition of titrant KOH and the blue shift was observed. These spectral changes were accompanied by five quasi-isosbestic points (at 235, 254, 274, 285, and 362 nm), which appear at high pH values (>11.5) and continue until the end of the titration. This suggests that equilibrium related to proton transfers may occur in this pH range. The exact number of equilibria that occurred in the investigated system can be obtained by plotting A-diagrams [33]. The corresponding A-diagram, presented in Figure 6B, showed three strictly linear sections for various wavelength combinations. The spectroscopic 2Cl3HP pH-dependency studies and the obtained UV results proved that three equilibria occurred in the studied 2Cl3HP aqueous system. It is worth noting that the first step in the deprotonation of 2Cl3HP is directly associated with its monocationic state as a consequence of the acidification of the 2Cl3HP pattern sample in the aqueous pH scale (0–14).

The relationships between the absorbance wavelength of 260 nm and pH values were plotted, and an example is presented in Figure 6B. The pKa values were calculated using the standard procedure based on the Henderson–Hasselbalch equation [34].

It has been proven that heterocyclic nitrogen is protonated by the action of an acid. However, this finding, supported by the experimental data (not included in this report but proved), did not refer in any way to the research systems studied here.

The data obtained from calculation with the model assumed (Table 3) in the CVEQUID program [35] were presented as adequate experimental dissociation constant values (pKa) of 2Cl3HP and shown in Table 4, together with the observed theoretical effects. The analysis of pKa values suggested that the data were determined with high accuracy. Moreover, these data correlate well, although they were obtained through independent measurements (spectrophotometric and potentiometric) with optimal but different conditions as well as different measurable parameters (absorbance and/or potential, respectively) of the studied 2Cl3HP system.

To resolve the debate related to the ionic forms of 2Cl3HP that were created during the studied protonation as well as deprotonation process, these forms were established and presented in Figure 7. Interestingly, the pKa values determined for 2Cl3HP were higher than those of 2HP. This indicates that the stronger basic character of 2Cl3HP is due to the presence of the Cl^−^ substituent in this molecule.

### 2.7. 2Cl3HP Affinity to CT-DNA

Electronic absorption titration was carried out by gradually increasing the concentration of CT-DNA from 0 to 113 μM at a constant concentration of the 2Cl3HP derivative (0.35 mM). Measurements were obtained using a computer-controlled automatic UV spectroscopic system to maintain constant time (3 min) between the addition of biomolecule titrant as well as product formation and stabilization of the solution (Figure 8). For quantitative comparison of the ability and strength of binding of 2Cl3HP with CT-DNA, the value of binding constant (K_b_) was calculated from the spectrophotometric data, based on changes in absorbance using Wolfe–Shimer Equation (3) [36] after the addition of different amounts of CT − DNA titrant solution:(3)$$\frac{\left[\mathrm{CT}-\mathrm{DNA}\right]}{\varepsilon_{a}-\varepsilon_{f}}=\frac{\left[\mathrm{CT}-\mathrm{DNA}\right]}{\varepsilon_{a}-\varepsilon_{f}}+\frac{1}{K_{b}\left(\varepsilon_{b}-\varepsilon_{f}\right)}$$
which included appropriate molar extinction coefficients (ε_a_ = A_obs_/[2Cl3HP]); ε_f_ determined by free pyrazine derivative spectrum with known concentration; ε_b_ established for 2Cl3HP-DNA adducts after the complete DNA binding.

To prove that the low-molecular, hydrophilic 2Cl3HP can interact with *CT*−DNA, in vitro investigation was performed to determine the binding constant value of 2Cl3HP, which described its adduct formation with *CT*−DNA.

The UV spectra of 2Cl3HP titrations by DNA were examined without pH value changes using biological Tris-HCl buffer (pH 7.40) as a solvent for preparing each pattern solution sample. It can be observed that the UV spectrum of 2Cl3HP exhibited two absorption bands in the absence of *CT*−DNA at a wavelength range of 200–400 nm. However, when the nucleic acid was added to the studied compound, significant changes were noted in its UV spectrum. Three types of isosbestic points were observed at 217, 253, and 290 nm, respectively. Their presence also proved that equilibria occurring in the studied system are related to the formation of the stable 2Cl3HP-DNA adduct. Moreover, the band intensity of free 2Cl3HP at 239 nm decreased and a maximum intensity at 244 nm was observed, which was remarkably enhanced with the redshift together with the hypochromic effect. These spectral changes confirmed that interactions of 2Cl3HP with CT-DNA occurred in the studied system. According to the literature, the intensity of an individual band of substrate depends on the size of molecules and the degree of electronic coupling among chromophores [37]. The dependence of [DNA]/(εa − εf) as a function of [DNA] was plotted to determine the actual binding constant value describing the specific type of interactions in the studied system (Figure 8B).

Interestingly, it is also possible to perform a comparative analysis of the two typical low-molecular binders: 2HP [1] and 2Cl3HP. Based on their values of the binding constant, it can be concluded that 2Cl3HP interacts with DNA almost 1.65-fold stronger compared to 2HP. Moreover, the stronger affinity of 2Cl3HP to the studied biomolecule (DNA) is certainly related to the chloride substituent present in its structure and the associated effect.

### 2.8. DNA Binding Studies

In this study, the molecular docking of 2Cl3HP, 2HP, and PZA with B-DNA was simulated in AutoDock 4.2 to determine the potential activity of compounds and molecular targets. PZA docking served as a benchmark due to its known activity and well-characterized mechanism of action as a drug. The predominant ligand–B-DNA complexes obtained by docking are shown in Figure 9, and the detailed interactions of the optimized structures along with significant distances are shown in Figure 9. By comparing the occurrence of populations, it can be unequivocally stated that the highest affinity for DNA is exhibited by 2Cl3HP (66% of the dockings). The predominant structure of 2HP and PZA complexes with DNA is slightly >50% and PZA differs only by 2%. It may also be important to consider the energy of the complex, which indicates the stability of the resulting system, and which can be discussed due to the comparability of the results obtained by molecular docking under the same computational conditions. The 2Cl3HP-DNA and PZA-DNA complexes have similar relative energy. The energy of the former (−4.12 kcal/mol) is lower than that of the latter (−3.16 kcal/mol) by only about 0.96 kcal/mol. Only the third index, the inhibition constant, calculated in the program, allows deciding which of these two ligands has a higher affinity for DNA. Compared to 2Cl3HP, 2HP has a much higher inhibition constant (4.82 mM). The experimentally determined values of the constants describing the interaction of both 2Cl3HP and 2HP with DNA were qualitatively consistent, with a higher value obtained for the chloride derivative.

This highlights the greater probability of binding of 2Cl3HP to DNA. Thus, using the obtained results, the investigated ligands can be ordered by decreasing DNA binding affinity as follows: 2Cl3HP > 2HP > PZA. Molecular docking is a cheap method for the preliminary estimation of noncovalent interaction. It was used in this case to obtain some statistics and judge whether such interaction plays an important role or not. What is worth noting is that the estimation based on docking overlaps with our experimental results, which in turn suggests the use of this low-cost method when starting considerations on the interaction with DNA.

All the investigation in this work shows that small molecules interact with B−DNA in the minor groove, as shown in Figure 9. The interaction site of the tested compounds is a characteristic cavity formed by three nucleotide pairs (GC, AT, AT), as shown in Figure 9. It is well known that the shape of the cavity must correspond with the ligands. In this case, a perfect covering between the cavity and the ligands was observed. All three investigated compounds were bound at the same site. The interactions between the molecules and the DNA groove were characterized as nonspecific π–π interactions occurring between two deoxyribose rings and a pyrazine ring inserted between them, in which the rings are arranged almost parallelly. The distance between the pyrazine and deoxyribose rings was similar in all cases, ranging from 3.5 to 4 Å. Additionally, after the ligand fitted into the cavity, the formation of nonspecific hydrogen bonds with phosphate was noted in the DNA chain. Hydrogen bonds were observed between the oxygen atom of phosphate and the external hydrogen atom of the substituent group for 2HP and PZA, while in the case of 2Cl3HP, one of the hydrogen bonds was formed between the nitrogen atom N(7) and the hydrogen atom from the –CH group in the DNA chain. The distances between the oxygen and nitrogen atoms involved in the formation of hydrogen bonds ranged from 2.7 to 2.8 Å, and that between the nitrogen and carbon atoms was 3.1 Å.

It can be found in the scientific reports that structure-based drug design has been used in the pharmaceutical industry for many years as a guiding approach to identify lead compounds and develop new therapeutic agents [38,39]. It has been established that nucleic acids are the molecular targets of many chemotherapeutic anticancer drugs in clinical use, which paved way for the design and development of drugs targeting nucleic acids for the treatment of genetic and acquired diseases. Much information about specific ligand–DNA or ligand–RNA binding sites can be found in structural databases. According to the study of Sheng [40], standard minor groove binders are polyamides or heterocyclic dications such as DAPI, Hoechst 33,258 (fluorescent stains), berenil (an anti-infective drug used for animals), dystamycin (cytotoxic and anticancer drug), and pentamidines (antimicrobial agent). These groove binders prefer AT-rich regions and were later further developed to extend their action of binding to other nucleotide pairs (TA, GC, and CG) [41,42]. The available reports indicate that a specific DNA-binding area should be determined for compounds with potential biological or therapeutic activity, or such compounds should be designed considering the possible interactions that can occur in this area. This study showed that the investigated ligands were similar to other active compounds in terms of interaction with the DNA.

### 2.9. Cytotoxic Activity

The cytotoxicity of 2Cl3HP toward human keratinocytes, HaCaT cells, was checked in this study. In our work, we aimed to confirm that ligand complexation enhances the cytotoxic effects of free ligands. We selected cell lines available in our laboratory, MCF-7 and HCT-116, and a non-cancerous HaCaT cell line, for this study. Since our compound exhibited low cytotoxic activity in comparison to clinically used chemotherapeutics, we did not extend the study to other cell lines. The activity of the compound was examined in the concentration range of 1–100 µM and after an incubation time of 72 h. The antiproliferative effects of 2Cl3HP were assessed using the MTT assay, which determines cell viability based on the conversion of the tetrazolium salt to formazan by mitochondrial dehydrogenase of metabolically active cells. The assay revealed that 2-chloro-3-hydrazinopyrazine did not exhibit cytotoxic activity toward HaCaT cells. At the examined concentration range of 0–50 µM, the viability of cells treated with 2-chloro-3-hydrazinopyrazine was 100% and at the highest concentration of 100 µM, cell viability was 80% (Figure 10). The cytotoxic activity of 2Cl3HP was in line with the results observed in our previous study on 2HP, which displayed comparable activity in the examined concentration range [1]. In addition, the research showed similar findings as observed in the case of PZA, which also did not display cytotoxic activity toward human hepatoma cells, HepG2 [43]. In a similar study, Singh et al. [44] showed that no cytotoxic activity toward HepG2 cells was observed at the highest examined PZA concentration of 50 mM.

### 2.10. Antimicrobial Susceptibility

Our previous studies showed the high antifungal activity of pyrazine derivatives at a low pH, supporting their potential application as drugs for the treatment of vaginal and gastric mycoses [45]. Due to the development of high bacterial and fungal resistance to the known used therapeutics, we designed new pyrazine derivatives as potential antimicrobial drugs [46,47,48]. The compound 2HP, whose structure and cytotoxicity was previously reported [1], and 2Cl3HP, which is the object of the current investigation, were examined for their antibacterial and antifungal activity. The obtained results confirmed that they have no antibacterial (Table 5) or antifungal (Table 6) activity, as indicated by the calculated MIC of >250 µM.

It was expected that decreasing the pH of the culture medium may favor the protonation of compounds and thus improve their transport to fungal cells, enhancing their biological activity. However, the obtained results did not show desired results [49,50]. The findings discussed herein seem to be surprising due to the structural similarities of both the studied compounds to pyrazine-2-carboxamide, which has been therapeutically used since the 1960s against Mycobacterium tuberculosis and is known commercially as bactericidal or bacteriostatic pyrazinamide. PZA is recognized as one of the top antituberculosis drugs in the world [51]. Interestingly, the pyrazine motif is generally defined as a desirable pharmacophore by the designers of microbial (pro)drugs. However, based on the previously reported MIC values of a series of pyrazine derivatives (PTCA, PZA, PAOX, 2HP, 2Cl3HP, ABMAP) which differ by the neighboring functional group (N(1) heteroaromatic atom), a remarkable conclusion can be drawn—the stimulation of antimicrobial activity is related to the substituent type located precisely at the second position of the 1,4-diazine element. The mechanism of PZA action proposed by Zhang et al. [52] and Mitchison et al. [53] suggests that the compound undergoes initial hydrolysis, resulting in the formation of an active metabolite—pyrazinoic acid. The acid present in M. tuberculosis causes a decrease in pH and turn induces abnormalities of membrane functions (e.g., energy disruption or transport inhibition). Interestingly, PZA could penetrate membranes through passive diffusion, and once inside the cell, the prodrug is deaminated by pyrazinamidase to pyrazinoic acid, which is released to the cytoplasm by an inefficient efflux system. It should be noted that some strains of M. tuberculosis have developed resistance toward PZA by a modification that allows them to produce pyrazinamidase.

The possible initiation reaction types that improve or stimulate the antimicrobial activity of PZA analogs should be evaluated for establishing the proper conditions and inducers for each of the studied pyrazine derivatives. The selected substituent can be activated by a specific, potentially individual reaction pathway without any analogy between the type of PZA activation or specific inducers dedicated to the functional group of the mentioned pyrazine derivatives. Moreover, in-depth knowledge of the mechanism underlying the antimicrobial action, as well as the established and selected stimulators of the individual functional substituent of pyrazine ring, can guide researchers to analyze and prove the antimicrobial action of the studied class of compounds through their pre-activation.

## 3. Materials and Methods

### 3.1. Chemicals

All starting materials were commercially available and purchased from Sigma Aldrich. The test compound with CAS number 63286-28-2 has a purity of 95%.

### 3.2. Spectrophotometric Investigation

All electron spectra were obtained using an Evolution 300 double-beam spectrophotometer (Thermo Fisher Scientific, Waltham, MA, USA). The cuvette holders were thermostated to maintain the constant temperature (25.0 ± 0.1 °C) of the solutions throughout the measuring time. The apparatus was also equipped with a magnetic stirrer and a computer-controlled automatic microtitrator (CerkoLab, Gdańsk, Poland). All spectrophotometric microtitration, which allowed for determining the acid-base properties of 2Cl3HP (0.25 mM), was performed in the wavelength range of 200–450 nm. The standard procedure of such measurements requires optimal conditions to be established. The sample was dissolved first in 1 mM HCl, and a 10 mM standardized KOH solution was used as a titrant. All titrations were performed in the pH range of 3.88–11.80. The deprotonation constant values of 2Cl3HP were determined using the Henderson–Hasselbalch equation imported into OriginPro software [54] and based on the theoretical model of dissociation reactions supported by the obtained experimental data. The spectrophotometric titrations were repeated three times to confirm the reproducibility of data registered for the complete deprotonation of 2Cl3HP.

### 3.3. DNA Interaction Studies

Ultraviolet (UV) microtitration, including the repetitions, was carried out in the tris-(hydroxymethyl)-amino methanation (Tris-HCl) buffer solution (consisting of 5 mM Tris-HCl and 50 mM NaCl, pH 7.40). Titrations that were automatically controlled by computer were done using the CerkoLab microinjector at 25 °C and 200–400 nm. Electronic absorption spectra were recorded after adding different amounts of each of the individual biomolecule solutions.

The concentration of freshly prepared calf thymus-DNA (CT-DNA) was calculated based on the absorbance measured at 260 nm and the calibration curve [εDNA 6600 (base pairs) M^−1^ cm^−1^] [55]. For a solution of CT-DNA (Sigma-Aldrich) in Tris-HCl, a ratio of UV absorbance of 1.8–1.9 at 260 and 280 nm was obtained, which indicated that the DNA was sufficiently free of protein (Appendix A). 2Cl3HP was dissolved in Tris-HCl to obtain a 25 mL solution with a mass concentration of 0.35 mM, which was used as an initial and pure pyrazine derivative sample.

### 3.4. Potentiometry

The titration system consisted of a titration cell, a magnetic stirrer, and an automatic microtitrator with Hamilton’s syringe having a volume of 1.0 mL and a step-up volume of 0.00835 mL. A 10.0 mM KOH (carbon dioxide-free) solution was used as a titrant. Potentiometric titrations were performed on the CERKO Lab system automatic titrator with the CERKO program, using InLab 423 combined glass–Ag/AgCl electrodes (Mettler-Toledo). The combined electrode was used after immediate calibration of its parameters (E0 = 382.16; S = −54.06). A constant temperature of 25 ± 0.1 °C was maintained throughout the measurement. The samples containing 2Cl3HP (0.25 mM) were dissolved in HCl (1 mM). The ionic strength of all working solutions was controlled using NaClO4 solution (10 mM). A 2.5 mL sample of the above-mentioned composition was used to register each potentiometric titration. The experimental data obtained from potentiometry were analyzed using the CVEQUID program [35], based on an algorithm that matches the assumed equilibria model, with the measurement data using the Gauss-Newton–Marquardt [56] iterative method to solve nonlinear problems. All potentiometric experiments were performed in the pH range of 1.42–11.03. The resolution of the voltage measurements was <0.1 mV. Three titrations were included simultaneously in calculations for the deprotonation of the studied 2Cl3HP solution.

### 3.5. Cell Culture and Cell Viability Assay

The human skin keratinocyte cell line HaCaT (Cell Line Services, Darmstadt, Germany) was cultured in a high-glucose Dulbecco’s Modified Eagle Medium (Sigma-Aldrich, Darmstadt, Germany), which was supplemented with 10% fetal bovine serum (Sigma-Aldrich, Darmstadt, Germany), 100 µg/mL streptomycin, and 100 units/mL penicillin (Sigma-Aldrich, Darmstadt, Germany). Culturing was carried out in a humidified atmosphere at 37 °C with 5% CO_2_. Cell viability was determined using the MTT (3,(4,5-dimethylthiazol-2-yl)-2,5-diphenyltetrazolium bromide) assay. Briefly, cells were seeded in 96-well plates at a density of 5 × 103 cells/well and treated with 2-chloro-3-hydrazinopyrazine at a concentration of 1–100 µM for 72 h. Subsequently, MTT (0.5 mg/mL) was added to the wells and the cells were incubated for 3 h at 37 °C. After incubation, the medium was discarded and the formazan crystals were dissolved in dimethyl sulfoxide (DMSO, 100 µL/well). The absorbance of the solution was measured at 550 nm using a plate reader (Victor, 1420 multilabel counter).

### 3.6. Statistical Analysis

Values are expressed as mean ± standard error (SE) of three independent experiments. Statistical analysis was performed using GraphPad Prism 5.0 (GraphPad Software) with one-way ANOVA [57], which is used to determine the statistical significance between the means of several treatment groups. One-way ANOVA with Tukey’s post hoc tests was used in this study to analyze the differences between control cells and cell treatments with various concentrations of 2-chloro-3-hydrazinopyrazine. A *p*-value of < 0.05 was considered statistically significant.

### 3.7. Antimicrobial Susceptibility Test

The antimicrobial activity of the studied compounds was determined using commercial bacterial Staphylococcus aureus (ATCC 25923), Bacillus cereus (PCM 2003), Escherichia coli (ATCC 25922), and fungal strains Candida albicans (ATCC 10231), Candida glabrata (DSM 11226), and Candida krusei (DSM 6128) by applying the broth microdilution method according to the guidelines of the Clinical and Laboratory Standards Institute (M07-A10 and M27-A3 documents). Briefly, the reference and tested compounds were dissolved in DMSO. Serial dilutions of compounds were prepared in 96-well microtiter plates using Mueller-Hinton broth and RPMI 1640 medium (Corning), which were adjusted to pH 4.0, 5.5, or 7.0 with hydrochloric acid. To each dilution in a ratio of 1:1, the inoculum amounting to 105 CFU/mL of all studied microorganisms (prepared from 24 h bacterial and fungal cultures grown at 37 °C) was added. The plates were incubated for 24 h at 37 °C, and the well with the lowest antimicrobial concentration, in which no growth of bacteria and fungi was observed, was considered as having minimum inhibitory concentration (MIC). Ciprofloxacin, levofloxacin, and fluconazole were used as control antimicrobial agents. Unless stated otherwise, all media and consumables were purchased from Sigma-Aldrich.

### 3.8. Computational Details

Calculations were performed using the Gaussian 09 software package [58], and results were obtained using the DFT approach. The analyses of structure, conformation, and tautomerism of the studied compounds were conducted with B3LYP (Becke’s three-parameter Lee–Yang–Parr) [59] functional and the split-valence 6-311+G** basis set in vacuo. It has been shown previously that the use of such a type of basis set reproduces experimental results reasonably well and is not expensive in terms of computational time [60]. The molecular structures of the compounds were fully optimized, and the total Gibbs free energies were determined at the same level of theory, along with vibration frequencies and their IR spectra. In addition, electronic absorption spectra (UV–Vis) of the optimized molecules were also calculated using the time-dependent density functional theory (TD-DFT) at the same level of theory.

Acid-base properties were studied using continuous two solvent models of water, to check which one is more suitable in this regard. All the gas-phase minima geometries were optimized using the PCM [61,62] and SMD [63] methods. The pKa values were determined based on the thermodynamic cycle described previously [29]. The free energy of the proton was assumed as −6.28 kcal/mol in a vacuum and −265.9 kcal/mol in water. The correction of the Gibbs free energy change resulting from the conversion of standard states (1 M to 1 atm) was also applied and was assumed to be 1.89 kcal/mol [30,60].

Partition coefficient studies were conducted in the water/n-octanol standard system. Gas-phase minima geometries were optimized at PCM/B3LYP/6-311+G** and SMD/M06-2X/6-311+G** using the solvent models of water and n-octanol, as described elsewhere [28]. The Gibbs free energies in water and n-octanol determined from these calculations were used further for calculating the logP value. All estimations were carried out at a temperature of 298.15 K and pressure of 1 atm (for the gas phase) or standard state 1 M (for the solvent model).

### 3.9. Molecular Docking

To predict the probable interaction sites of PZA, 2HP, and 2Cl3HP with the B-DNA, fragment molecular docking was used. The ligand structures were created using the MOLDEN program and then optimized in Gaussian 09 software [58] until all eigenvalues of the Hessian matrix were positive. The structures of the three B-DNA fragments were extracted from Protein Data Bank and used for molecular docking. Molecular docking simulations for the binding of PZA, 2HP, and 2Cl3HP to the B-DNA fragment were carried out in the AutoDock 4.0 program [64]. Lamarckian Genetic Algorithm [65] was used for a conformational search of the space around DNA. A docking space of 26 × 32 × 45 Å^3^ was declared to cover the entire DNA model. For docking simulation, the following parameters were applied: 300 genetic algorithm runs and 27,000 generations in each run, 3,500,000 was the maximum number of energy evaluations per docking run, 2 torsional degrees of freedom, and 1000 were external grid energy. The grid maps had a spacing of 0.375 Å, which is a default value. All other parameters were default settings. The ligand was initially placed at a random point in the declared docking space, and after the simulation, the most populated clusters were selected for further analysis.

## 4. Conclusions

Our studies have shown that pyrazine derivatives, especially those with a hydrazine substituent, can be potentially developed as nontoxic therapeutics for the treatment of unmet diseases in the future. Both the hydrazinopyrazines studied in this work were low-molecular compounds with the ability to interact with DNA—the most commonly recognized critical cellular target in chemical carcinogenesis.

Based on the presented properties of the investigated compound in an aqueous solution, such as pKa or the logP value, its biological activity could be probable. pKa allows us to determine the presence of acid-base equilibria and the preferred ionic forms, and logP identifies lipophilicity, thus allowing us to determine the possibility of penetration into cells or interaction with other particles in the environment. An important element was also the analysis of the interaction with DNA, resulting in quite a strong interaction in the characteristic cavity of the double-strand—it has a stronger effect than the drug pyrazinamide. Therefore, such a characterization is a significant precedent of biological tests, which finally confirms or denies the activity of the compound. The analyzed properties of 2Cl3HP indicate its potentially possible activity, which was contradicted by the tests performed on living cells and microorganisms. However, the described properties and the indicated possibilities of interaction with DNA that allow us to suspect the lack of activity towards microorganisms, whether due to its present ionic form or the problem with penetrating the living cell, are impossible to observe. This is a clue for further research on the modification of the form of administration of the compound.

Some slight differences were found in the relative Gibbs free energies of the isomers A-C (Rys. 2), which proved the significant participation of all forms in the structure. Moreover, the analyses confirmed the existence of four acid-base equilibria in the solution and the hydrophilic nature of the studied compound—2Cl3HP. A small groove formed by a system of three nucleotide pairs (GC, AT, AT) was found in the docking study as the most probable interaction site of 2HP, 2Cl3HP, and PZA with DNA. These interactions were identified as specific binding as well as nonspecific hydrophobic and hydrogen bonding. Furthermore, 2Cl3HP showed the highest affinity for DNA among the tested compounds, which was unequivocally confirmed by both experimental and theoretical analyses.

## Figures and Tables

**Figure 1 molecules-27-03704-f001:**
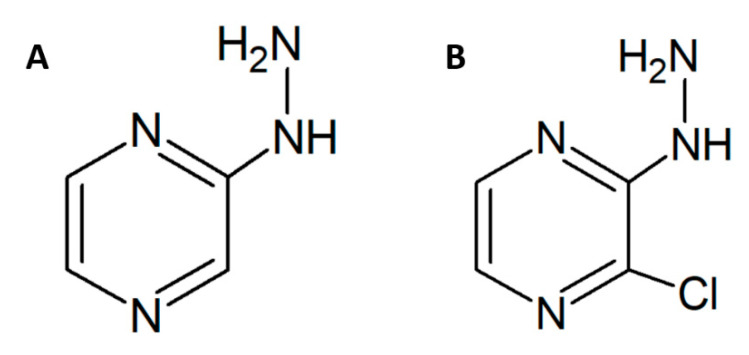
The structure of the 2-hydrazinopyrazine (*2HP*)—(**A**), and 2-chloro-3-hydrazinopyrazine (*2Cl3HP*)—(**B**).

**Figure 2 molecules-27-03704-f002:**
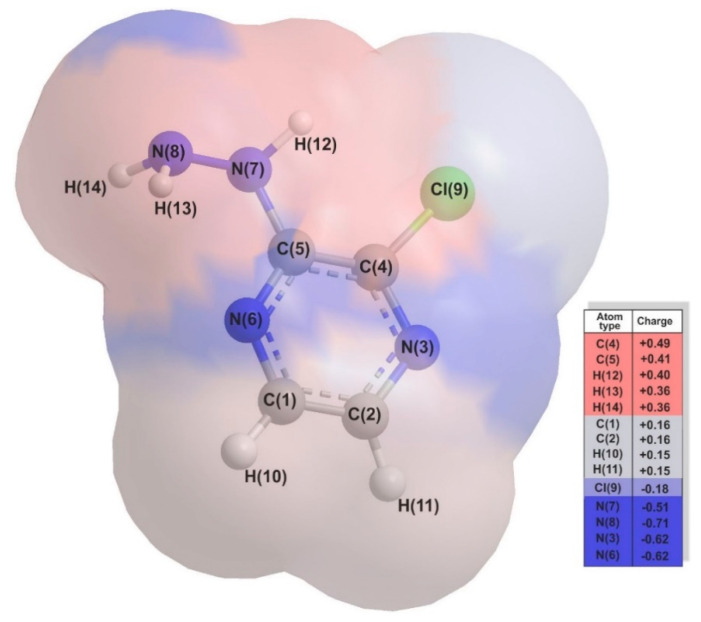
The structure of the 2−chloro−3−hydrazinopyrazine (2Cl3HP) together with the spatial map of the electron density distribution (Mulliken charges) received in the Chem3D^®^ program by *Perkin Elmer*.

**Figure 3 molecules-27-03704-f003:**
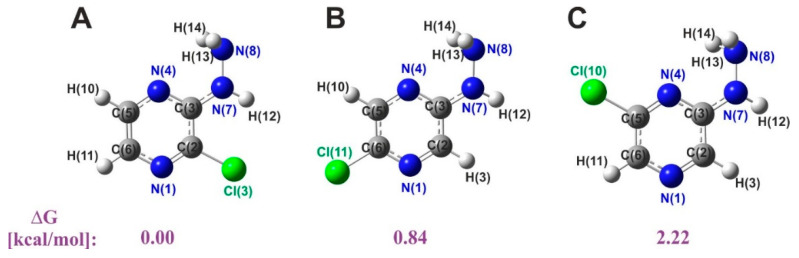
Theoretically obtained optimized isomer structures of chlorohydrazinopyrazine (ClHP) with atom numbering. Their relative Gibbs free energies in kcal/mol: (**A**) ΔG: 0.00, (**B**) ΔG: 0.84, (**C**) ΔG: 2.22, respectively. Calculations were performed with the use of B3LYP/6-311+G**. Relative values were compared with the energy value of the A isomer (by subtracting them) and converting from a.u. (Hartree/particle) to kcal/mol using a multiplier of 627.5095 kcal·mol^−1^·a.u.^−1^.

**Figure 4 molecules-27-03704-f004:**
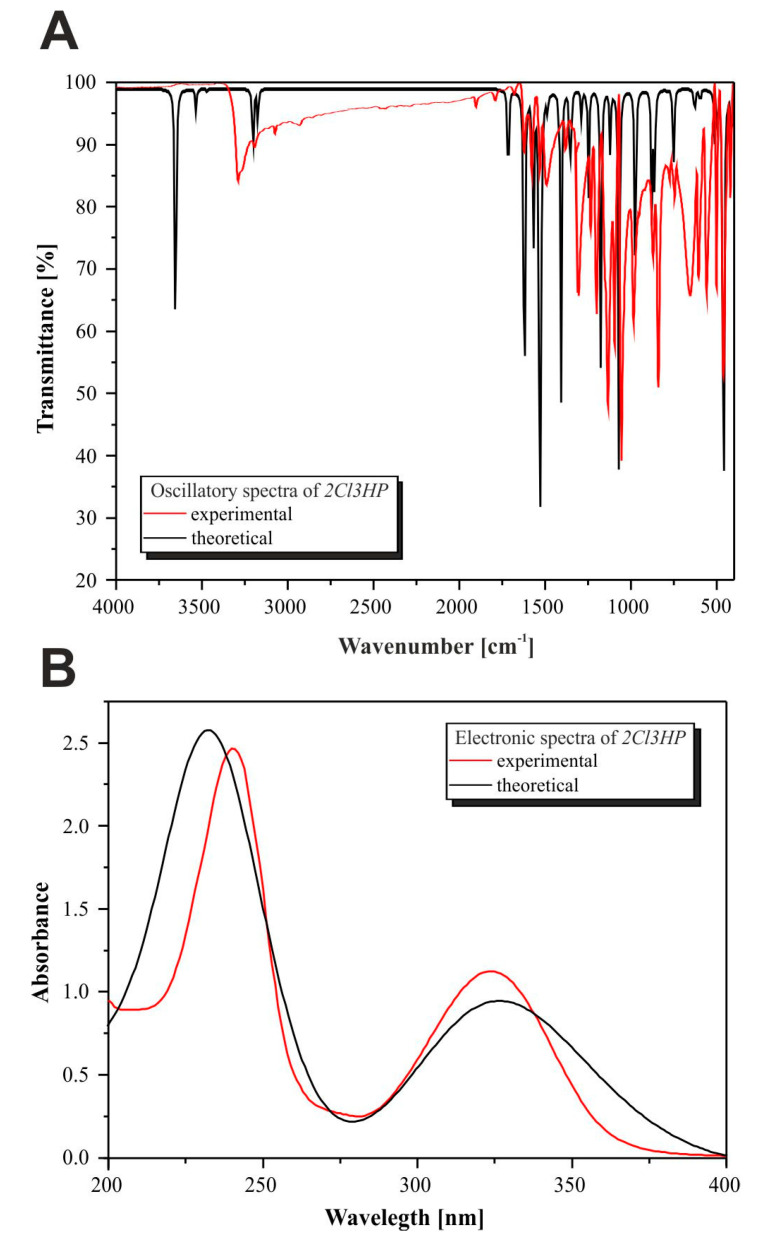
(**A**) Experimental IR spectra—red line, and for comparison computed at B3LYP/6-311+G** in vacuo—black line. (**B**) Experimental UV-Vis spectra—red line, and for comparison computed at TD-DFT/B3LYP/6-311+G** in vacuo.

**Figure 5 molecules-27-03704-f005:**
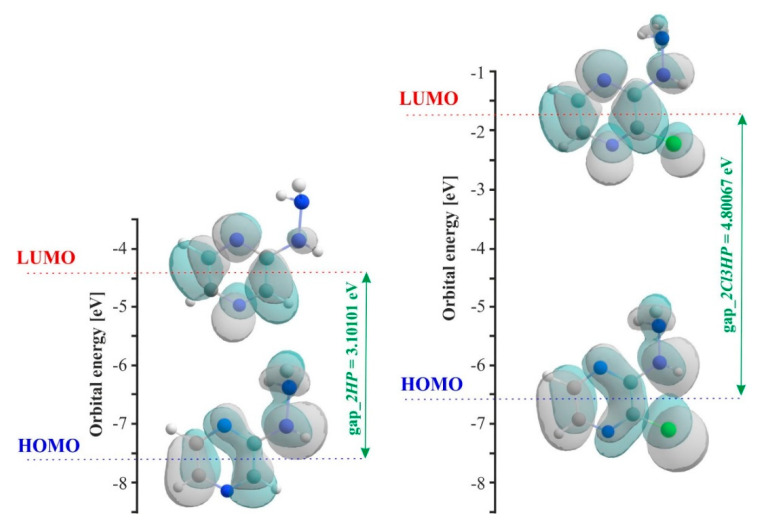
Molecular orbital diagram projected of HOMO−LUMO structures for *2HP* and *2Cl3HP* calculated at B3LYP/G-311+G** in vacuo.

**Figure 6 molecules-27-03704-f006:**
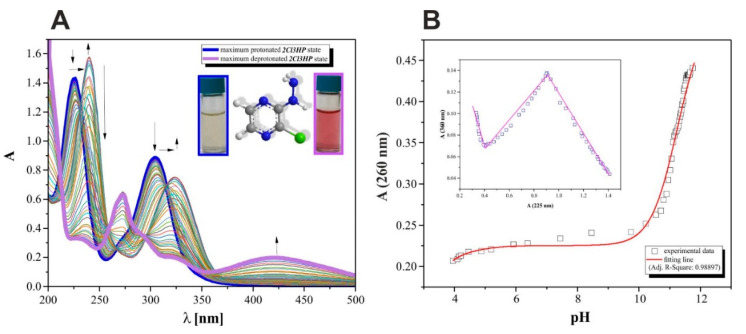
Spectrophotometric results: (**A**) set of UV spectral curves of 2Cl3HP (0.25 mM) sample acidified as a function of pH (3.88–11.80); (**B**) absorption at 260 nm vs. measured pH; experimental data were presented by squares; calculation results were included as red line; R^2^ = 0.98897.

**Figure 7 molecules-27-03704-f007:**
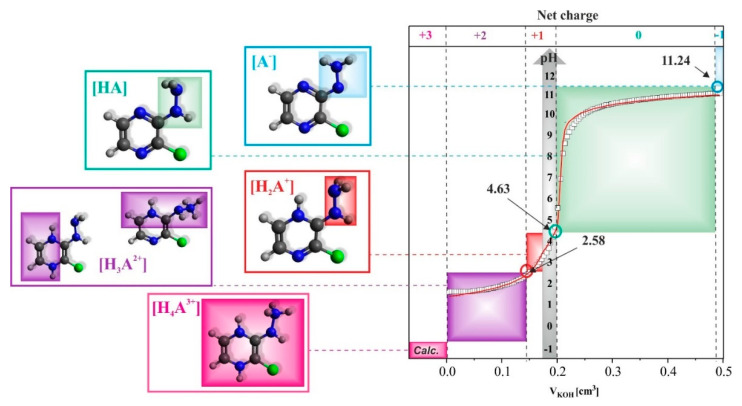
The ionization states as net charge established for the 2Cl3HP based on the potentiometric, spectroscopic as well as calculated data received (left); potentiometric titration data (squares) obtained during KOH (10 mM) volume additions to 0.25 mM 2Cl3HP probe dissolved in 1 mM HCl/0.01 M NaClO_4_ together with the red line as a result of CVEQUID calculations (right).

**Figure 8 molecules-27-03704-f008:**
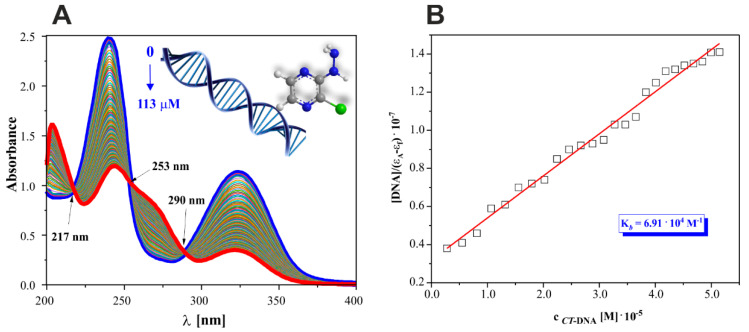
Absorption titration of 2Cl3HP derivative (**A**) with increasing concentrations of *CT*−DNA (0−113 µM). Arrows show isosbestic points obtained upon increasing *CT*−DNA concentration; (**B**) the plot of [DNA]/(ε_A_ − ε_f_) vs. [DNA]; □, experimental data points at 239 nm for 25 portions of DNA added; solid line, linear fitting of the data (R^2^ = 0.98785).

**Figure 9 molecules-27-03704-f009:**
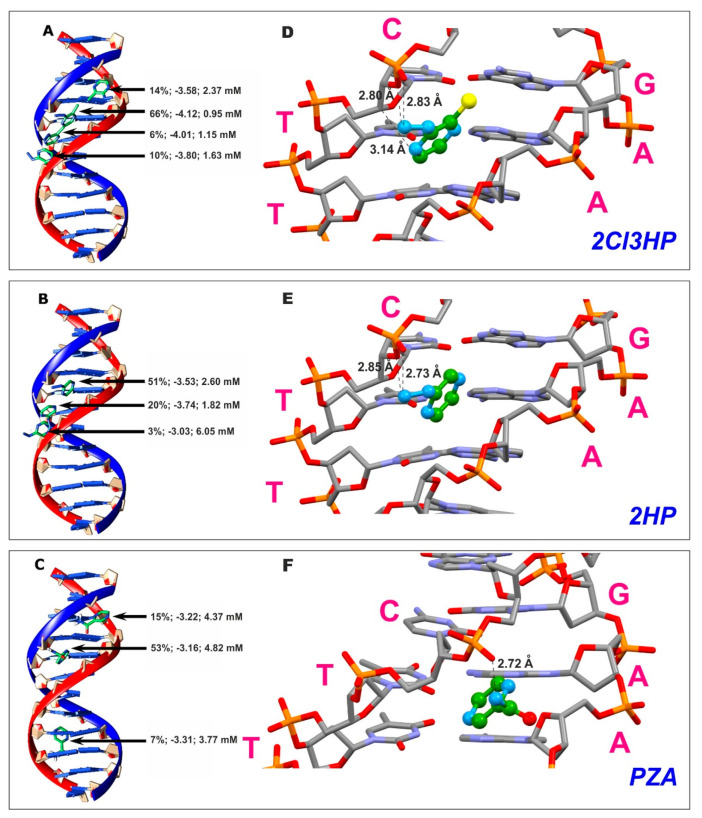
Molecular docked model of complexes (**A**) 2Cl3HP, (**B**) 2HP, and (**C**) PZA with B−DNA. For each received model, the population prevalence [%], Gibbs free energy [kcal/mol], and the inhibition constant [mM] as calculated by AutoDock 4.2 were given. The detailed interactions of the optimized structures (**D**) 2Cl3HP, (**E**) 2HP, and (**F**) PZA with B−DNA along with significant distances [Å]; the purple symbols mean: **C** is cytosine, **G** is guanine, **T** is thymine and **A** is adenine nucleobases. The following color scheme was used for the ligands: green for a carbon atom, blue for a nitrogen atom, red for an oxygen atom.

**Figure 10 molecules-27-03704-f010:**
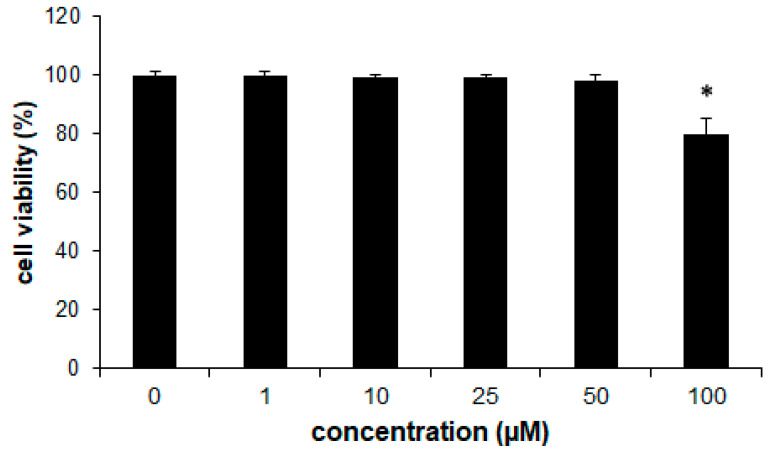
The effects of 2-chloro-3-hydrazinopyrazine toward HaCaT cells. The viability of HaCaT cells was determined with the MTT assay after 72 h. Values represent mean ± SE of three independent experiments. *p* < 0.05 (*) indicates differences between control and treated cells.

**Table 1 molecules-27-03704-t001:** Comparison analysis of hydrazinopyrazine derivatives with computed HOMO and LUMO energies, energy gaps, and other parameters in eV computed in vacuo at B3LYP/6-311+G** level of theory.

No.	Energy (eV)	2HP	2Cl3HP
1	HOMO (ground state)	−7.58 [1]	−6.58
2	LUMO (ground state)	−4.48 [1]	−1.78
3	Energy gap = HOMO − LUMO	3.10 [1]	4.80
4	Ionization energy	7.58	6.58
5	Electron affinity	4.48	1.78
6	Hardness	3.10	2.40
7	Softness	0.32	0.42
8	Chemical potential	−6.03	−4.18

**Table 2 molecules-27-03704-t002:** Gibbs free energies (G) in water and n-octanol were obtained as a result of DFT simulations and on their basis, determined partition coefficient (logP) values.

Method	$G_{H_{2}O}\;(\mathrm{Hartree}/\mathrm{Part}.)$	$G_{n-octanol}\;(\mathrm{Hartree}/\mathrm{Part}.)$	logP
SMD/M06-2X/6-311+G**	−834.675525	−834.499315	−0.428
PCM/B3LYP/6-311+G**	−834.674420	−834.500245	−0.508

**Table 3 molecules-27-03704-t003:** Proposed deprotonation equilibria model used for aqueous systems studied in calculations (potentiometry and spectrophotometry) and DFT approach; 2Cl3HP neutral form was assigned as a [HA].

Conjugated Ionic Forms of 2Cl3HP Involved in Equilibrium	Assumed Model of Protolytic Reaction	Deprotonation Constant (Acidity)
trication/dication	H_4_A^3+^ + OH^−^ ⇆ H_3_A^2+^ + H_2_O	pKa_1_
dication/monocation	H_3_A^2+^ + OH^−^ ⇆ H_2_A^+^ + H_2_O	pKa_2_
monocation/neutral	H_2_A^+^ + OH^−^ ⇆ HA + H_2_O	pKa_3_
neutral/monoanion	HA + OH^−^ ⇆ A^−^ + H_2_O	pKa_4_
water formation	H_3_O^+^ + OH^−^ ⇆ 2 H_2_O	pK_aq_

**Table 4 molecules-27-03704-t004:** The values of dissociation constants obtained experimentally **A**. for protonated *2Cl3HP* determined at 25 °C in an aqueous medium from *n* titrations (I = 0.01 M NaClO_4_; *n* ≥ 3); standard deviations computed thereby refer to random errors only. For comparison pKa values obtained by DFT approach **B**.

Method	pKa Values			
	pKa_1_	pKa_2_	pKa_3_	pKa_4_
A. Experimental Part				
Potentiometry	ND *	2.58 ± 0.08	4.63 ± 0.02	ND *
UV-spectroscopy	ND *	2.98 ± 0.04	4.99 ± 0.23	11.24 ± 0.06
B. Theoretical Part				
PCM/B3LYP/6-311+G**	−33.05	−20.36	−3.15	19.66
SMD/M06-2X/6-311+G**	−18.48	−17.77	−2.04	19.69

ND *—not detected.

**Table 5 molecules-27-03704-t005:** The antibacterial activity of investigated compounds is defined as MIC [µM]. Ciprofloxacin and levofloxacin were used as references.

Compound	Bacteria Strain		
	Gram (+)		Gram (−)
	*Bacillus cereus*	*Staphylococcus aureus*	*Escherichia coli*
**Ciprofloxacin**	0.25	0.02	0.005
**Levofloxacin**	0.25	0.02	0.01
**2HP**	>250	>250	>250
**2Cl3HP**	>250	>250	>250

**Table 6 molecules-27-03704-t006:** The antifungal activity of investigated compounds is defined as MIC [µM]. Fluconazole was used as a reference.

Strain	pH	2HP	2Cl3HP	Fluconazole
** *C. albicans* **	7.0	>250	>250	125
	5.5	>250	>250	>250
	4.0	>250	>250	>250
** *C. glabrata* **	7.0	>250	>250	62.5
	5.5	>250	>250	>250
	4.0	>250	>250	>250
** *C. krusei* **	7.0	>250	>250	62.5
	5.5	>250	>250	250
	4.0	>250	>250	250

## Data Availability

Not applicable.

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
