# Peer review of "Low-Molecular Pyrazine-Based DNA Binders: Physicochemical and Antimicrobial Properties"

_molecules, 2022, doi:10.3390/molecules27123704_

Round 1
Reviewer 1 Report
The manuscript starts from the huge review about mechanisms of the affinity of DNA and connected with them mechanisms of antimicrobial and antifungal activity of some pyrazine derivatives. After that, the analysis of spectral properties and calculated Gibbs energies, as well as electronic properties is given. I have some remarks for the presentation and analysis of these data, but they do not influence on my decision. The results connected with the main goal with the research are the spectral investigation of DNA complexing with the pyrazine derivatives. But only in the end of the manuscript, a reader finds out that the substances studied are absolutely safe not only for human body but to bacteria and fungi. The main conclusion that the approach presented in the manuscript (manly based on the fact that DNA may form complexes with pyrazine derivatives studied) can help to find new biologically active substances looks doubtful and unproven. That is why I suppose that the results themselves could be published with another title, introduction, and conclusion.
Author Response
Reviewer #1
The manuscript starts from the huge review of mechanisms of the affinity of DNA and connected with them mechanisms of antimicrobial and antifungal activity of some pyrazine derivatives. After that, the analysis of spectral properties and calculated Gibbs energies, as well as electronic properties is given. I have some remarks for the presentation and analysis of these data, but they do not influence on my decision.
The results connected with the main goal with the research are the spectral investigation of DNA complexing with the pyrazine derivatives. But only in the end of the manuscript, a reader finds out that the substances studied are absolutely safe not only for human body but to bacteria and fungi. The main conclusion that the approach presented in the manuscript (manly based on the fact that DNA may form complexes with pyrazine derivatives studied) can help to find new biologically active substances looks doubtful and unproven. That is why I suppose that the results themselves could be published with another title, introduction, and conclusion.
Authors Answer for Reviewer#1
The Authors changed the title as well as redrafted chapters questionable for Reviewer#1. The interaction possibility of low molecular compounds like 2HP or 2Cl3HP with biomolecules like DNA is worth noting as a cellular target during investigations. Even safe compounds like the popular anti-tuberculosis pyrazine drug – PZA have to be activated at lower pH in the initial step and then also modified inside bacteria to show high antimicrobial activity. Due to the above, the known ionic form (pKa values) of the compound studied with penetrating biological barriers property (logP) and which could interact with biomolecules like DNA (Kb values) seems to be crucial data to create the possibility of developing those subject areas. We decided to show the preliminary studies results to indicate some interactions, ionization analogy together with some differences in energy gap value between the standard PZA, published 2HP, and main object 2Cl3HP, respectively. The preparation of the complete comparison analysis was appropriate in our opinion to find relevant like described pKa, Kb and logP values in the article reviewed.
Indeed, the main considered goal of our work was achieved, which was to study the interaction of the tested compounds with DNA as a potential molecular target of many pyrazine-based drugs. Three selected compounds were compared: 2Cl3HP, 2HP, PZA as a reference point. We successfully combined theoretical and experimental research. The obtained results are consistent. While implementing our main goal, we came to a surprising conclusion, which suggests that 2Cl3HP is the compound that binds the most strongly to DNA. Interestingly, the specific area of binding to DNA was determined, as well as the specificity of the cavity and the non-specificity of the interactions.
Taking into account the very limited availability of the description of this compound in the literature, we aimed at its precise description. Searching the conformational hypersurface using the DFT approach, computed, and experimental spectra - all allowed for the analysis of the structure. Additionally, the HOMO LUMO orbitals approach allowed to describe the electronic properties. Then, the use of calculations using the water model in conjunction with potentiometry and spectrophotometry made it possible to learn about the properties of 2Cl3HP in solution: acid-base properties and lipophilicity. Anything to get to know the compound whose potential activity was a signpost for us.
Additionally, the reviewed paper was recommended by another journal exactly to the theoretical section of Molecules due to the main part of the manuscript is related to calculations. Although, both the experimental and theoretical parts were positively received by the reviewers, one Reviewer recommended choosing the theoretical section as definitely more appropriate for this article. Therefore, we decided to send the work to the section with a calculation profile.
Reviewer 2 Report
In this manuscript, the structure of chlorohydrazinopyrazine, its physico-chemical and cytotoxic properties, and site and nature of interaction with DNA were studied by both experimental and theoretical studies.
There are several problems with the manuscript:
- The title: In the manuscript, only three isomers of chlorohydrazinopyrazine were studied, substituent effect in the title is not proper.
- The calculation method and basis sets should be careful consideration. B3LYP/6-311+G(d, p) maybe enough for geometry optimization and TDDFT calculation. In my opinion, a more accurate calculated level is needed for of ionization energy and electron affinity calculations. Moreover, pKa values were calculated at PCM/B3LYP/6-311+G** and SMD/M06-2X/6-311+G** levels, both of them give the different results compared to the experiment. What’s the origin of the different results? Solvent model or Density functionals?
- Net charge: Which kind of charge did the author used?
- Supporting Information: The significant digits are too much for the bond lengths and bond angles.
Thus, I cannot recommend a publication of the manuscript in its present form. A publication might be possible after correction.
Author Response
Reviewer #2
There are several problems with the manuscript:
- The title: In the manuscript, only three isomers of chlorohydrazinopyrazine were studied, substituent effect in the title is not proper.
The Authors changed the title of the manuscript reviewed according to Reviewer#2 suggestion.
2. The calculation method and basis sets should be careful consideration. B3LYP/6-311+G(d, p) maybe enough for geometry optimization and TDDFT calculation. In my opinion, a more accurate calculated level is needed for of ionization energy and electron affinity calculations. Moreover, pKa values were calculated at PCM/B3LYP/6-311+G** and SMD/M06-2X/6-311+G** levels, both of them give the different results compared to the experiment. What’s the origin of the different results? Solvent model or Density functionals?
To improve the manuscript reported experimental as well as theoretical results (both supported each other), we included the additional elaborated UV spectra for 2HP and 2Cl3HP to check the propriety of the energy gap values calculated by using B3LYP/6-311+G(d, p) to those obtained from electronic spectra. The comparison analysis shows a very good agreement between them. We agree with Reviewer#2 that the specific pKa values calculated do not strictly correspond to those obtained from the experiment, however, the tendency is proper. Based on our earlier report with the same 2HP study types, we tried to receive adequate data to prepare some compilations about the two objects studied.
The reason for the different numerical values of the solvation energy is both solvation models and calculation methods. For example, the differences between energies from both approaches differ by more than 1.5 kcal/mol [R1]. This proofs that that different functional is the origin of energy differences. Similarly, the two models of the PCM and SMD solution differ in their assumptions. In the PCM method, the total energy of the solvation is the sum of the three components: electrostatic, dispersion-repulsion, and cavitation. PCM has limitations where non-electrostatic effects dominate the solution-related interactions. The SMD model also belongs to the continuum solvation models and is based on the quantum mechanical charge density of a solute molecule interacting with a Continuum Description of the Solvent. As stated in the paper [R2]: “The model separates the observable solvation free energy into two main components. The first component is the bulk electrostatic contribution arising from a self-consistent reaction field treatment that involves the solution of the nonhomogeneous Poisson equation for electrostatics in terms of the integral- equation-formalism polarizable continuum model (IEF-PCM). The cavities for the bulk electrostatic calculation are defined by superpositions of nuclear-centered spheres. The second component is called the cavity-dispersion- solvent-structure term and is the contribution arising from short-range interactions between the solute and solvent molecules in the first solvation shell. This contribution is a sum of terms that are proportional (with geometry-dependent proportionality constants called atomic surface tensions) to the solvent-accessible surface areas of the individual atoms of the solute.”
References mentioned and cited above:
[R1] V. S. Bryantsev, M. S. Diallo, A. C. T. van Duin, W. A. Goddard, Evaluation of B3LYP, X3LYP, and M06-Class Density Functionals for Predicting the Binding Energies of Neutral, Protonated, and Deprotonated Water Clusters, J. Chem. Theory Comput. 2009, 5, 4, 1016–1026.
[R2] A.V. Marenich, Ch.J. Cramer, D.G. Truhlar, Universal Solvation Model Based on Solute Electron Density and on a Continuum Model of the Solvent Defined by the Bulk Dielectric Constant and Atomic Surface Tensions, J. Phys. Chem. B 2009, 113, 18, 6378–6396.
3. Net charge: Which kind of charge did the author used?
We used the net charge provided by Mulliken. An adequate description was added in the Figure 1 caption.
4. Supporting Information: The significant digits are too much for the bond lengths and bond angles.
The data in Tables S1 have unified according to Reviewer#2 suggestion No. 4. and the number of significant digits seems to be proper now.
LIST OF CHANGES:
- The values in the Tables were unified (for energy as well as bonds or angles).
- The text of the manuscript was rewritten in some parts like an introduction.
- The title was changed to be more appropriate.
- The additional Figures about the energy gap values obtained experimentally were added (Figures S4 and S5) in Supporting Material.
Reviewer 3 Report
DFT calculations and experimental (potentiometric and spectrophotometric) analysis were combined to understand the structure of chlorohydrazinopyrazine, its physicochemical and cytotoxic properties, and site and nature of interaction with DNA. The main conclusion is that 2-chloro-3-hydrazinopyrazine (2Cl3HP) displayed the highest affinity to DNA; moreover, the compound did not exhibit toxicity toward human dermal keratinocyte.
I’m unfamiliar with the experimental study mentioned in this manuscript. My main concern is the motivation of DFT calculations. It seems that the theoretical DFT calculations have not relationship with the final conclusion, which is obtained from the experimental study. More discussions are required to explain the introduction of DFT calculations. It seems that this paper is not appropriate for the section of Computational and Theoretical Chemistry in MOLECULE. It can be submitted to another section that focus on experimental study.
Another minor concern is that significance digit in Table 1 should be unified. The number of significant digit for energy in EV can be two.
Author Response
Reviewer #3
I’m unfamiliar with the experimental study mentioned in this manuscript. My main concern is the motivation of DFT calculations. It seems that the theoretical DFT calculations have not relationship with the final conclusion, which is obtained from the experimental study. More discussions are required to explain the introduction of DFT calculations. It seems that this paper is not appropriate for the section of Computational and Theoretical Chemistry in MOLECULE. It can be submitted to another section that focus on experimental study.
Authors comment to Reviewer#3
We would like to thank Reviewer#3 for his/her suggestion. However, the reviewed paper was recommended by another journal exactly to the theoretical section of Molecules due to the main part of the manuscript is related to calculations. Although, both the experimental and theoretical parts were positively received by the reviewers, one Reviewer recommended choosing the theoretical section as definitely more appropriate for this article. Therefore, we decided to send the work to the section with a calculation profile.
Another minor concern is that significance digit in Table 1 should be unified. The number of significant digit for energy in EV can be two.
Authors comment to Reviewer #3
The data in Table 1 were unified according to the Reviewer#3 suggestion and the number of significant digits for energy is now two.
LIST OF CHANGES:
- The values in the Tables were unified (for energy as well as bonds or angles).
- The text of the manuscript was rewritten in some parts like an introduction.
- The title was changed to be more appropriate.
- The additional Figures about the energy gap values obtained experimentally were added (Figures S4 and S5) in Supporting Material.
Round 2
Reviewer 1 Report
After the corrections, the manuscript can be accepted in present form. I have not any remarks now.
Author Response
We would like to thank Reviewer#1 for His/Her positive decision.
Reviewer 3 Report
The author try to answer my question about the role of DFT calculations in the manuscript, they stated that "However, the reviewed paper was recommended by another journal exactly to the theoretical section of Molecules due to the main part of the manuscript is related to calculations. Although, both the experimental and theoretical parts were positively received by the reviewers, one Reviewer recommended choosing the theoretical section as definitely more appropriate for this article. Therefore, we decided to send the work to the section with a calculation profile"
This reply is not satisfactory. I do not agree with the reviewer. The revised manuscript does not show any improvement for the relationship between experimental results and theoretical study.
Maybe the editor select another reviewer to judge whether this paper is appropriate for this Journal.
Author Response
Our previous answer to the Reviewer’s #3 question was not fortunate. We apologize for that. The main goal of our work was to study the interaction of pyrazine derivative-based drugs with DNA as their molecular target. We used theoretical methods to support our experiments and also to show some tendencies qualitatively. We did not mean to develop any theoretical methods in this regard. We are aware, that our DFT results might not fit well in a quantitative way, but allow us to analyze for example conformational hypersurface, IR spectra, relative interaction energy, and finally HOMO LUMO orbitals to describe the electronic properties. Moreover, the use of calculations with the water model in conjunction with potentiometry and spectrophotometry made it possible to learn about the properties of 2Cl3HP in solution: acid-base properties and lipophilicity.
This manuscript is a resubmission of an earlier submission. The following is a list of the peer review reports and author responses from that submission.
Round 1
Reviewer 1 Report
Title: Low-molecular pyrazine-based DNA binders: substituent effect vs. interactions and antimicrobial properties
Authors: Paulina Mech-Warda , Artur Giełdoń , Anna Kawiak , Natalia Maciejewska , Mateusz Olszewski , Mariusz Makowski , Agnieszka Chylewska
My comments to this manuscript are as follows:
- Third paragraph of the introduction section is written without the support of relevant reference(s).
- The abbreviations of a compound or a term should be given at their first appearance in the manuscript.
- Since this study is related to the derivative of 2-hydrazinopyrazine its structure along with its derivatives, 2-chloro-3-hydrazinopyrazine (already given in the manuscript), should also be given.
- Give more details about the theoretically obtained isomers of 2-chloro-3-hydrazinopyrazine for instance what are their exact names, the methodology for obtaining them and how the energy difference was calculated. How many isomers are stable?
- It is also recommended to discuss the frontier molecular orbitals of the various isomers of 2-chloro-3-hydrazinopyrazine.
- Lines 207-210: please provide suitable reference.
- The study of the UV spectrum of DNA should be used to examine the binding mode of DNA with ligands. Unfortunately, authors studied the UV spectra of 2-chloro-3-hydrazinopyrazine, because the change in the spectral characteristics of ligand after binding with DNA via the intermolecular forces certainly occur. So, I suggest authors add the study of UV spectra of DNA with the addition of 2-chloro-3-hydrazinopyrazine. Further, they should use difference spectra for the calculation of various parameters because in the selected wavelength range DNA also absorbs considerably. It is also not clear in Figure 7 (B) that these data-points are taken from which wavelength.
- The molecular docking study, without additional experimental support, is highly speculative. The authors provide no real reasoning for why they believe that the compounds bind at minor groove. Solid experimental support should be provided in addition to the computational method.
- The comparison of the properties of 2-chloro-3-hydrazinopyrazine with its parent compound 2-hydrazinopyrazine should be done thoroughly and not at only selected instances.
Reviewer 2 Report
The manuscript ijms-1646933 devoted the actual field of the medicinal chemistry, namely design low-molecular pyrazine-based DNA binders and can be interested to the specialists working in this field. The author’s opinion is clear and based on a wide range of recent publications. I am personally impressed by the structure of the article, the systematization of scientific data and the sequence of its presentation. The paper fit the Journal scope and formal requirements. However, it needs major revision before publication.
To improve the quality and perception of the manuscript I would suggest paying attention to following comments:
.
- The introduction to the article does not sufficiently substantiate the need and novelty of the research. The author must argue why pyrazine derivatives were chosen as objects of study. It is necessary to analyze the current literature on the biological activity of these heterocycles. In general, the introduction should be more focused on the propersties of low-molecular pyrazines.Given the specifics of the journal, the authors should provide structural formulas of the studied derivatives.
- The origin of the tested compounds is unclear. If the authors synthesized them, it is necessary to provide the methods of synthesis and the corresponding analytical characteristics. If they are commercial compounds, it is necessary to present their source of origin (supplier, purity, etc).
- Given the specifics of the paper in the discussion and conclusions, it is necessary to provide opportunities for further use of the results, especially in the context of drug design and medicinal chemistry.
- Moderate English changes required. There are grammar and orthographical errors in the manuscript, which should be corrected
My decision is major revision.
Author Response
Reviewer 2:
Comments and Suggestions for Authors
The manuscript ijms-1646933 devoted the actual field of the medicinal chemistry, namely design low-molecular pyrazine-based DNA binders and can be interested to the specialists working in this field. The author’s opinion is clear and based on a wide range of recent publications. I am personally impressed by the structure of the article, the systematization of scientific data and the sequence of its presentation. The paper fit the Journal scope and formal requirements. However, it needs major revision before publication.
To improve the quality and perception of the manuscript I would suggest paying attention to following comments:
- The introduction to the article does not sufficiently substantiate the need and novelty of the research. The author must argue why pyrazine derivatives were chosen as objects of study. It is necessary to analyze the current literature on the biological activity of these heterocycles. In general, the introduction should be more focused on the properties of low-molecular pyrazines. Given the specifics of the journal, the authors should provide structural formulas of the studied derivatives.
The introduction has been supplemented with additional literature references describing low molecular weight pyrazine derivatives. The structures of the described 2-hydrazinopyrazine and 2-chloro-3-hydrazinopyrazine derivatives were added to the manuscript (as Figure 1).
2. The origin of the tested compounds is unclear. If the authors synthesized them, it is necessary to provide the methods of synthesis and the corresponding analytical characteristics. If they are commercial compounds, it is necessary to present their source of origin (supplier, purity, etc).
In the experimental section, necessary information about the commercial origin of the compound was added. Page number 17 and lines 547-548 of the revised manuscript.
3. Given the specifics of the paper in the discussion and conclusions, it is necessary to provide opportunities for further use of the results, especially in the context of drug design and medicinal chemistry.
The described studies constitute an initial analysis of the physicochemical properties that are key for active biological compounds. The performed microbiological tests are also very basic research. To propose the use of test compounds, a series of more advanced studies would have to be performed. It is also possible to consider changes to make the compounds more active (for example by modifying ionizable groups) or to consider the possible catalytic activity. However, based on the research carried out so far, without wanting to overestimate the potential of a chemical compound, we would not like to speculate as far-reaching conclusions as to the use of the substance.
4. Moderate English changes required. There are grammar and orthographical errors in the manuscript, which should be corrected.
The manuscript has been corrected for grammar and orthographical errors, English has been improved.
List of changes:
- Text fragments have been added in the introduction.
- Literature references have been added in the introduction.
- The missing abbreviation of the test compound name was added in the introduction, and additionally, abbreviations were added when the names were first used in the Results and discussion section.
- Fragments have been added to the introduction text.
- A fragment of the text in the first paragraph of the Results and discussion section has been changed.
- Figure 1. The structure of the 2-hydrazinopyrazine (2HP) and 2-chloro-3-hydrazinopyrazine (2Cl3HP) was added.
- Figures have been renumbered.
- The signature of Figure 3 (in the new numbering) has been made more detailed by the method of calculating the value.
- A reference has been added in section 2.4.
- References have been renumbered.
- Subsection 3.1 has been added to Chapter 3, Materials and Methods, which describes the origin of the test compound.
- Subsections of chapter 3 have been renumbered.
- The wavelength (239 nm) was added upon Figure 8B (new number).
Reviewer 3 Report
The manuscript by Chylewska and co-workers describes the pyrazine compound's affinity to DNA and its substituent effect/antimicrobial properties. From B3LYP/6-311+G** basis set, authors found that 2-Chloro-3-hydrazinopyrazine showed the highest affinity to DNA, and from cytotoxicity studies by using HaCaT cells it did not show any toxicity. Authors also performed UV spectra analysis and found that 2-Chloro-3-hydrazinopyrazine interacts with DNA almost 1.65-fold stronger than 2-hydrazinopyrazine. The decreasing order of DNA binding was found to be 2Cl3HP>2HP>PZA. Authors also performed antimicrobial/antifungal activity of these investigated compounds but unfortunately, none of the compounds showed any activity from the MIC of >250uM assay
Overall, this is an interesting manuscript; I recommend its publication after addressing the below-mentioned comments.
- For pKa values: 6-31++G** is a better basis set and improves pKa estimations. (Ref: Phys. Chem. A 2016, 120, 28,5726-5735). Could you kindly check with these basis set and see is there any difference?
- For better pKa predictions, wB97Xd functional will be helpful
- Authors can consider computing NBO analysis to find out the occupancy of loan pair at N (population analysis) in the presence of Cl. This will help in supporting the interaction of lone pair at N to DNA(ref: https://doi.org/10.1016/j.comptc.2020.113095)
- There is no control for the cytotoxicity experiments it is always better to compare to known compounds whether it is performing a better activity or not?
- Why didn't the authors use the SMD model with B3LYP functional to do pKa calculations? SMD models give better predictions over PCM (Ref: Phys. Chem. A2016, 120, 28, 5726–5735)
- Just curious to know whether the authors try to find the binding of DNA to 2Cl3HP from the Isothermal Titration calorimetry?
Author Response
Reviewer 3:
The manuscript by Chylewska and co-workers describes the pyrazine compound's affinity to DNA and its substituent effect/antimicrobial properties. From B3LYP/6-311+G** basis set, authors found that 2-Chloro-3-hydrazinopyrazine showed the highest affinity to DNA, and from cytotoxicity studies by using HaCaT cells it did not show any toxicity. Authors also performed UV spectra analysis and found that 2-Chloro-3-hydrazinopyrazine interacts with DNA almost 1.65-fold stronger than 2-hydrazinopyrazine. The decreasing order of DNA binding was found to be 2Cl3HP>2HP>PZA. Authors also performed antimicrobial/antifungal activity of these investigated compounds but unfortunately, none of the compounds showed any activity from the MIC of >250uM assay
Overall, this is an interesting manuscript; I recommend its publication after addressing the below-mentioned comments.
Thank you for your comments on the results of our paper. Three out of them refer to the theoretical methods in our paper. We would like to answer them together.
- For pKa values: 6-31++G** is a better basis set and improves pKa estimations. (Ref: Phys. Chem. A 2016, 120, 28,5726-5735). Could you kindly check with these basis set and see is there any difference?
- For better pKa predictions, wB97Xd functional will be helpful.
- Why didn't the authors use the SMD model with B3LYP functional to do pKa calculations? SMD models give better predictions over PCM (Ref: Phys. Chem. A2016, 120, 28, 5726–5735)
We would like to thank the reviewer for her/his comments. But we cannot agree in this regard. Based on our both experience and knowledge there is no good, better, or best method for pKa calculations. The only thing we could agree on is the fact that the 6-31++G** basis set and wB97Xd functional should give in theory more precise results. The main difference is that wB97Xd describes both short and long-distance effects well, while B3LYP only short. wB97XD does not take into account dispersion contribution (empirically) and B3LYP does not. The energy expression in wB97XD is the sum of the Kohn-Sham energy and the dispersion term calculated between each pair of atoms. To get better results, one can use B3LYP with Grimme’s dispersion instead of wB97XD, so this does not necessarily lead to better results. As far as scaling is concerned, the computation time for these functions in B97XD/b3lyp is ~1. 3. In theory, it is necessary to consider dispersion contribution in systems where they are significant (now the most difficult question arises: and when are they significant?).
But for example, the problem of pKa predictions is not only connected with the basis set. One could find in a literature well-fitted theoretical from semiempirical, quantum chemical pKa values with the experimental ones. In our recent paper (J. Phys. Chem. A 2020, 124, 538−551) we calculated pKa values for a series of pyridine, its N-oxide, and their derivatives. In our calculations, we used a direct thermodynamic cycle involving the gas phase Gibbs free energy calculations (at three levels of theory, M06-2X, B3LYP, and G4MP2) and the PCM solvation model. Calculations were performed for four scale factors alpha values (1.1, 1.2, 1.3, and 1.4) in two solvents, water, and acetonitrile. Furthermore, our studies were enhanced with the pKa calculations involving the Muckerman’s et al. solvation Gibbs free energy correction (Biochim. Biophys. Acta, Bioenerg. 2013, 1827, 882−891). Our previous studies on pyridine and its derivatives have shown a strong influence of the scale factor-alpha on pKa values. In the case of pKa values for the set of pyridines, referred to the water environment, the best results (the lowest RMSD) were obtained for scale factor-alpha equal to 1.1, regardless of the level of the gas phase calculation used. In turn, the lowest RMSD was observed for scale factors alpha equal to 1.2 in acetonitrile. For the pyridine N-oxides studied, both in water and acetonitrile, the lowest RMSD was found for the scale factors alpha equal 1.2. In the above-mentioned calculations, we used very simple PCM.
In the present work, we focused only to show the tendency and assess the values within the studied compound. In this case, we used the model and method which is a compromise between the computational time and predictability. Some time ago we came to terms that there is no simple theoretical approach for pKa calculations. This is presumably caused by the fact that the solvent models do not reproduce well all solvent properties in theory.
Here we used the SMD solvent model in conjunction with the M06-2X functional, these are well-coordinated methods, and the model was parameterized in the group of Truhlar’s group, who also developed the functional. The tests were performed using the PCM model and the SMD model to compare the results obtained with both approaches - the results turned out to be qualitatively consistent. In the case of the described studies, it cannot be clearly stated which results are more qualitatively consistent. There are no available literature data providing pKa or logP values, and the experimental results described in the paper did not make it possible. Fortunately, they support our experimental results very well and allow us to draw reasonable conclusions.
4. Authors can consider computing NBO analysis to find out the occupancy of loan pair at N (population analysis) in the presence of Cl. This will help in supporting the interaction of lone pair at N to DNA(ref: https://doi.org/10.1016/j.comptc.2020.113095)
In this work, based on experimental research, an ionic form was found at a pH of approx. 7 - it is a neutral form of compounds and the DNA interaction tests were performed and described in the paper based on this form. In our opinion, such an analysis would not bring any new conclusion. But we think that it is worth considering such calculations in our separate paper in the future detailed work with the analysis of interactions with DNA. In this paper, we present several experimental techniques supported by theoretical calculations. This paper is long enough with supporting information by itself. This is also one of the reasons why these calculations should be the subject of another paper.
5. There is no control for the cytotoxicity experiments it is always better to compare to known compounds whether it is performing a better activity or not?
Based on the results obtained, no demonstrated cytotoxicity was found at any level, therefore, a more in-depth comparative analysis was abandoned.
6. Just curious to know whether the authors try to find the binding of DNA to 2Cl3HP from the Isothermal Titration calorimetry?
As a preliminary experimental test in confrontation with theoretical calculations (molecular docking), spectrophotometric titration was performed, the results of which are discussed in the paper. More advanced experimental studies have not been carried out so far. We will certainly consider conducting such an experiment in our further research.
List of changes:
- Text fragments have been added in the introduction.
- Literature references have been added in the introduction.
- The missing abbreviation of the test compound name was added in the introduction, and additionally, abbreviations were added when the names were first used in the Results and discussion section.
- Fragments have been added to the introduction text.
- A fragment of the text in the first paragraph of the Results and discussion section has been changed.
- Figure 1. The structure of the 2-hydrazinopyrazine (2HP) and 2-chloro-3-hydrazinopyrazine (2Cl3HP) was added.
- Figures have been renumbered.
- The signature of Figure 3 (in the new numbering) has been made more detailed by the method of calculating the value.
- A reference has been added in section 2.4.
- References have been renumbered.
- Subsection 3.1 has been added to Chapter 3, Materials and Methods, which describes the origin of the test compound.
- Subsections of chapter 3 have been renumbered.
- The wavelength (239 nm) was added upon Figure 8B (new numbered).
Reviewer 4 Report
At the beginning, it should be emphasized that the work belongs to theoretical rather than experimental work, therefore, in my opinion, it should be submitted to a journal where the attention is mainly focused on theoretical considerations. Perhaps the more appropriate journal for this manuscript is JMolStruct. Considering that biological research concerns only two derivatives that do not show significant antitumor or antibacterial activity, it is difficult to conclude that the presented results are very important for a scientific achievement. Therefore, I do not recommend this manuscript for publication in this journal.
Author Response
Reviewer 4:
Comments and Suggestions for Authors
At the beginning, it should be emphasized that the work belongs to theoretical rather than experimental work, therefore, in my opinion, it should be submitted to a journal where the attention is mainly focused on theoretical considerations. Perhaps the more appropriate journal for this manuscript is JMolStruct. Considering that biological research concerns only two derivatives that do not show significant antitumor or antibacterial activity, it is difficult to conclude that the presented results are very important for scientific achievement. Therefore, I do not recommend this manuscript for publication in this journal.
We cannot discuss this in this regard because any impression is personal. This paper is an experimental one supported with theoretical calculations. There is a lot of such type of papers published in the Int. J. Mol. Sci. Below, You can find some examples of articles published in the Int. J. Mol. Sci. with the results of studies directly related to our propositions of data included in the reviewed paper:
https://doi.org/10.3390/ijms22062807
https://doi.org/10.3390/ijms20246308
https://doi.org/10.3390/ijms18051069
https://doi.org/10.3390/ijms20246308
Taking into account the thematic scope of the journal and the works published in it, we believe that our multi-method work based on the use of theoretical and experimental methods with the use of cytotoxicity and microbiological tests is appropriate. In addition, it is worth presenting the obtained results, despite the low promising results in terms of biological activity, since there is no description of the properties of this derivative in the literature.
List of changes:
- Text fragments have been added in the introduction.
- Literature references have been added in the introduction.
- The missing abbreviation of the test compound name was added in the introduction, and additionally, abbreviations were added when the names were first used in the Results and discussion section.
- Fragments have been added to the introduction text.
- A fragment of the text in the first paragraph of the Results and discussion section has been changed.
- Figure 1. The structure of the 2-hydrazinopyrazine (2HP) and 2-chloro-3-hydrazinopyrazine (2Cl3HP) was added.
- Figures have been renumbered.
- The signature of Figure 3 (in the new numbering) has been made more detailed by the method of calculating the value.
- A reference has been added in section 2.4.
- References have been renumbered.
- Subsection 3.1 has been added to Chapter 3, Materials and Methods, which describes the origin of the test compound.
- Subsections of chapter 3 have been renumbered.
- The wavelength (239 nm) was added upon Figure 8B (new numbered).
Round 2
Reviewer 2 Report
The authors took into account all the comments and improved the article. The paper maz be accepted.
Reviewer 4 Report
The compounds described in the manuscript are biologically inactive, therefore the theoretical studies that dominate are of little value. I reiterate my earlier position that this work should be published in a journal focused more on theoretical research. In my opinion, the presented results are of low scientific value, they do not bring any significant information into the development of new biologically active structures. What is the point of theoretical calculations if the compounds are not biologically active? What contributes to science IR spectra computed at B3LYP / 6-311 + G ** level of theory for 2-chloro-3- hydrazinepyrazine, 2-chloro-5-hydrazinepyrazine, and 2-chloro-6- hydrazinepyrazine since the compounds are well known?